# Bone Healing in Rat Segmental Femur Defects with Graphene-PCL-Coated Borate-Based Bioactive Glass Scaffolds

**DOI:** 10.3390/polym14183898

**Published:** 2022-09-18

**Authors:** Ozgur Basal, Ozlem Ozmen, Aylin M. Deliormanlı

**Affiliations:** 1Department of Orthopedics and Traumatology, Emsey International Hospital, Pendik, İstanbul 34912, Turkey; 2Department of Pathology, Faculty of Veterinary Medicine, Burdur Mehmet Akif Ersoy University, Istiklal Yerleskesi, Burdur 15030, Turkey; 3Department of Metallurgical and Materials Engineering, Manisa Celal Bayar University, Yunusemre, Manisa 45140, Turkey

**Keywords:** bioactive glass, graphene, osteogenesis, rat femur defect, in vivo, bone healing

## Abstract

Bone is a continually regenerating tissue with the ability to heal after fractures, though healing significant damage requires intensive surgical treatment. In this study, borate-based 13-93B3 bioactive glass scaffolds were prepared though polymer foam replication and coated with a graphene-containing poly (ε-caprolactone) (PCL) layer to support bone repair and regeneration. The effects of graphene concentration (1, 3, 5, 10 wt%) on the healing of rat segmental femur defects were investigated in vivo using male Sprague–Dawley rats. Radiographic imaging, histopathological and immuno-histochemical (bone morphogenetic protein (BMP-2), smooth muscle actin (SMA), and alkaline phosphatase (ALP) examinations were performed 4 and 8 weeks after implantation. Results showed that after 8 weeks, both cartilage and bone formation were observed in all animal groups. Bone growth was significant starting from the 1 wt% graphene-coated bioactive glass-implanted group, and the highest amount of bone formation was seen in the group containing 10 wt% graphene (*p* < 0.001). Additionally, the presence of graphene nanoplatelets enhanced BMP-2, SMA and ALP levels compared to bare bioactive glass scaffolds. It was concluded that pristine graphene-coated bioactive glass scaffolds improve bone formation in rat femur defects.

## 1. Introduction

The treatment of long bone defects resulting from trauma, tumor resections, congenital malformations and osteomyelitis are the most demanding bone pathologies. In cases of failure even after many serial surgeries, the extremity may need to be amputated [1]. To create optimal biological healing conditions in the shortest period, there is a tremendous need for porous biocompatible materials that can provide new bone regeneration and mechanical strength [2,3,4,5,6].

Bioactive glasses (BGs) are emerging materials that can be used to regenerate and heal both bone and skin [3,7]. BGs exhibit osteoconductive, osteoinductive, and osteo-stimulative characteristics, and are degradable in physiological fluids [8,9]. Some examples of osteoinductive growth factors that promote new bone formation and regeneration are bone morphogenetic proteins (BMP), vascular endothelial growth factor (VEGF), epidermal growth factor (EGF), and transforming growth factor (TGF) [10,11,12,13,14]. Unfortunately, synthesis is costly and growth factors’ extensive utility is limited by a low stability in scaffolds. Instead, osteoinductive nanomaterials like graphene nanoplatelets should be incorporated into the architecture of synthetic scaffolds.

One essential element that can enhance osteogenesis is boron, by inducing the osteogenic differentiation-marker gene synthesis during the proliferation and differentiation phase of BMSCs [7,8]. According to an in vitro study conducted by Gu et al. [9], borate-based 13-93B3 bioactive glasses showed better bone-healing performance in rat calvarial defect models compared to silicate-based 13-93 bioactive glass scaffolds [9]. Besides the advantages of boron-based bioactive glass scaffolds in bone regeneration, they have been reported to feature poor mechanical properties compared to silicate-based glasses [10]. Therefore, some biopolymers, such as poly (caprolactone) and poly (L-lactide-co-glycolide), and some inorganic nanomaterials, such as graphene, have been added to the bioactive glass structure to improve its mechanical properties [11,12,13].

In the past, graphene-containing PCL-coated 13-93B3 bioactive glass scaffolds were prepared by Türk and Deliormanlı [14]. An in vitro cytotoxicity analysis (XTT) showed that the graphene-containing scaffolds were not cytotoxic to pre-osteoblast MC3T3-E1 cells, and cell viability rates were higher compared to a control group after 7 days of incubation [14].

It was previously reported that graphene-containing, grid-like silicate-based bioactive glass-based, and graphene-containing PCL-coated composite scaffolds fabricated by robocasting had no detrimental effect on bone marrow mesenchymal stem cells in vitro [15]. Stem cells implanted onto these composite scaffolds fixed well to the surface and proliferated efficiently. In the absence of transforming growth factors, cells implanted on the scaffolds showed osteogenic differentiation [15].

Wang et al. found in their study that mesoporous bioactive glass-graphene oxide scaffolds had better cytocompatibility and osteogenesis differentiation with rat bone marrow mesenchymal stem cells than bare mesoporous bioactive glass scaffolds [16]. Furthermore, these bioactive scaffolds stimulated vascular ingrowth and promoted bone repair at the lesion site in rat cranial defect models. Similarly, Gao et al. [17] adopted graphene in combination with 58S bioactive glasses for bone tissue engineering using scaffolds fabricated by selective laser sintering. Human MG63 cells adhered to and proliferated on the surface of graphene-containing scaffolds, according to cell culture studies [18].

The potential of graphene derivatives, including some functional groups such as graphene oxide and reduced graphene oxide, for stem cell differentiation into osteogenic, chondrogenic, adipogenic, and neurogenic types has been investigated previously [19]. Functional groups of graphene equivalents are responsible for hydrophobic and electrostatic interactions with proteins which promote osteogenic differentiation. Recently, graphene-containing porous and oriented PCL scaffolds have been used in the regeneration of large osteochondral defects [20], enhancing fibrous, chondroid and osseous tissue regeneration. Additionally, the expressions of bone morphogenetic protein-2, collagen-1, vascular endothelial growth factor and alkaline phosphatase expressions were more prominent in PCL implanted groups in the presence of graphene [20]. However, despite the data available, the contribution of borate-based 13-93B3 scaffolds containing pristine graphene nanoplatelets to bone healing in vivo has not been reported yet.

The hypothesis of this study is that graphene-containing PCL-coated 13-93B3 porous scaffolds are effective bioactive composites in the treatment of segmental bone defects, and are good alternatives to autologous bone grafts. In this study, the effects of borate-based, 13-93B3 scaffolds coated with PCL containing pristine graphene nanopowders at different concentrations on bone healing in segmental bone defect model are investigated.

## 2. Experimental Studies

### 2.1. Materials

In the study, melt-derived borate-based 13-93B3 bioactive glass powders (5.5 Na_2_O, 11.1 K_2_O, 4.6 MgO, 18.5 CaO, 3.7 P_2_O_5_, 56.6 B_2_O_3_ wt%) having a median diameter of ~10 µm were used. PCL (Mw 80,000 g/mol) was purchased from Sigma-Aldrich, Darmstadt, Germany. Graphene nanopowders in platelet form were received from Graphene Laboratories Inc. Calverton, NY, USA. The platelets were 60 nanometers thick with particle sizes ranging from 3 to 7 μm, according to the manufacturer.

### 2.2. Scaffold Preparation

A polymer foam replication technique was used to fabricate porous bioactive glass-based scaffolds [12]. In the presence of 4 wt% ethyl cellulose, a suspension containing borate-based bioactive glass particles (40 vol%) and ethanol was prepared. Scaffolds were then made using cylindirical polyurethane foams (8.5 mm in diameter) as a template. The foams’ surfaces were covered using a dip-coating procedure that involved immersion in a bioactive glass suspension. The coated foams were then dried at 25 °C for 48 h before being subjected to a controlled heat treatment at 1 °C/min to degrade the polymeric foam up to 450 °C. They were then sintered for 1 h at 570 °C with a heating rate of 5 °C/min in an air atmosphere [11].

A poly (caprolactone) solution in di-chloromethane was stirred at 25 °C for 4 h at 5 wt%. The graphene nanopowders (at 1, 3, 5, or 10% wt%) were then added to the PCL solution and mixed for 1 h with a magnetic stirrer before homogenization for 15 min with a Bandelin Sonopuls (Bandalin, Berlin, Germany) ultrasonic probe [14]. A graphene-containing polycaprolactone solution was used to coat porous bioactive glass scaffolds using the dip-coating method. Graphene and PCL-coated scaffolds were then dried at room temperature for at least 48 h before characterization. The prepared 0, 1, 3, 5 and 10 wt% graphene nanoparticle-containing PCL-coated composite scaffolds were designated as BG, 1G-P-BG, 3G-P-BG, 5G-P-BG and 10G-P-BG, respectively.

The microstructure of these manufactured scaffolds was examined using an optical and scanning electron microscope (SEM, Zeiss, Gemini 500, Oberkochen, Germany), and a Fourier transform infrared spectrometer with an attenuated total reflectance module (FTIR-ATR, Thermo Scientific, Nicolet, IS20, Waltham, MA, USA) in the range of 525 cm^−1^ to 4000 cm^−1^. An XRD analysis of the bare borate-based bioactive glass scaffolds was made using a Panalytical Empyrean diffractometer (Malvern Panalytical BV, Brighton, UK) with Cu Kα radiation (λ = 1.5406 Å) in the range 10° to 90° (2θ). The scaffolds’ porosity was determined using the Archimedes method. Prior to animal implantation, the scaffolds were soaked in an ethanol bed overnight and then sterilized under UV light for 2 h.

### 2.3. In Vivo Experiments

All investigations were carried out in line with the Ministry of Health of Turkey, the Declaration of Helsinki, and the National Institutes of Health (NIH) of the United States’ Guide for the Care and Use of Laboratory Animals. The Experimental Animal Center and Ethics Committee of Burdur Mehmet Akif Ersoy University (MAKU) approved all experimental procedures in this study (Ethics number: 531, 17 July 2019).

#### 2.3.1. Rat Segmental Defect Model

Male Sprague–Dawley rats, from twenty to twenty-six weeks of age, 260–380 g, were purchased from the Experimental Animal Production and Experimental Research Center of Burdur Mehmet Akif Ersoy University, Turkey. Five groups of 5 specimens each were formed. A right femur segmental defect model was then created for all animals. Rats were fed and watered on a regular basis, and housed in individual cages in controlled rooms with 12-h light/dark cycles at 22 °C and 50% humidity. Prophylactic antibiotic (4 mg/kg gentamicin im.) was administered to all rats approximately 60 min before surgery. 100 mg/kg Alfamine 10% (Alfasan, Woerden, Holland) and 10 mg/kg Xlazinbio 2% (Bioveta, Czech Republic) were injected intraperitoneally to allow for a 30-min anesthetic period. Under anesthesia, the lateral thigh was shaved and a local antiseptic was applied. Animals were positioned on their left side and a 3 cm incision was made using a lateral longitudinal approach from mid-shaft to the distal femur. The incision was extended using a lateral parapatellar arthrotomy approach and a medial dislocation of the patella to gain access to the femoral notch, where a blunt dissection was made to the vastus lateralis muscle. After the right femur shaft was exposed, a 5 mm wide bone cut was performed with a mini saw and a bone segment removed. A prepared scaffold with a hole of 20 Ga in the middle was placed in the femur defect. The graft material was fixed with a 20 Ga- 4 cm retrograde intramedullary pin which was placed from intercondylar notch of the distal femur (Figure 1).

Rats implanted with bioactive glass-based composite scaffolds with codes BG, 1G-P-BG, 3G-P-BG, 5G-P-BG and 10G-P-BG, were designated as Control Group, Group 1, Group 2, Group 3 and Group 4, respectively. Exclusion criterion is the occurrence of surgical site infection. Two rats from each group were sacrificed at random four weeks after surgery, and three other rats were sacrificed eight weeks after surgery.

#### 2.3.2. Histopathological Method

All rats were anesthetized and euthanized at the end of the study. Bone samples were fixed in 10% neutral-buffering formalin for histological and immuno-histochemical studies during the necropsy. After a 2-day fixation period, bone samples were decalcified for two weeks using a 0.1 M ethylenediaminetetraacetic acid (EDTA) solution, tissues were processed and embedded in paraffin using an automatic tissue processor (Leica ASP300S, Wetzlar, Germany). Using a rotary microtome, five-micron serial sections were obtained from the paraffin blocks (Leica RM 2155; Leica Microsystems, Wetzlar, Germany). A light microscope was used to analyze one slice of each rat stained with hematoxylin and eosin (HE) and one section stained with the Picro Sirius Red method for collagen by a ready to use kit (ab150681, Abcam, Cambridge, UK).

For new bone formation (NFB; mm^2^) and the residual material area, histomorphometric variables were calculated (RMA; mm^2^). At 400× magnification, osteoblasts and osteoclasts were detected in a 1.23 mm^2^ area [21]. The entire faulty area was investigated in two dimensions and calculations were carried out in five distinct parts of each segment. A statistical analysis on the mean values of each group’s results was performed. An expert pathologist from a different institution who was unaware of the study design analyzed histopathological changes blindly.

#### 2.3.3. Immunohistochemistry Method

BMP-2 (against BMP-2 antibody (655229.111) (ab6285)), smooth muscle actin (Anti-alpha smooth muscle Actin antibody ((4A4); ab119952)), and ALP (Anti-ALP antibody; (ab67228)) were immuno-stained with the streptavidin biotin method for immuno-histochemical investigation. Abcam provided all primary antibodies and secondary kits (Cambridge, UK). For 60 min, the sections were treated with primary antibodies. A biotinylated secondary antibody and a streptavidin–alkaline phosphatase combination were used for immuno-histochemistry. The secondary antibody was the EXPOSE Mouse and Rabbit Specific HRP/DAB Detection IHC kit (ab80436) Abcam, Cambridge, UK. Antigens were identified using a chromogen diaminobenzidine (DAB). Instead of primary antibodies, an antibody dilution solution was used as a negative control.

As analyses were carried out in a blind approach, the immune-positivity of all slides was evaluated semi-quantitatively by taking staining intensity into account (0, absence of staining; 1, slight; 2, medium and 3, marked). The results of the image analyzer were then subjected to statistical analyses. The Database Manual Cell Sens Life Science Imaging Software System (Olympus Co., Tokyo, Japan) was used to perform morphometric studies.

#### 2.3.4. Statistical Analysis

SPSS version 23.0 was used to conduct statistical analysis (IBM, Armonk, NY, USA), and the Duncan test for independent samples was used to compare groups using a one-way analysis of variance (ANOVA). Statistical significance was defined as *p*-values of less than 0.001. G power 3.1 (Düsseldorf, Germany) software was used to calculate the sample size of two animals per group; sample size planning revealed an effect size d of 10.5 and an actual power of 0.98.

## 3. Results

### 3.1. Bioactive Glass Scaffolds

Figure 2a shows digital images of the bioactive glass-based PCL-coated composite scaffolds containing graphene nanopowders at different concentrations. Accordingly, scaffolds have a porous structure and graphene addition did not alter the morphological features significantly. The diameter and thickness of the fabricated, disc-shape scaffolds were measured at ~7 and ~3 mm, respectively. Optical microscope images shown in Figure 2b reveal the existence of macropores in the range of 300–500 μm. Graphene nanopowders homogeneously distributed within the PCL matrix cover the surface of the bioactive glass samples. Figure 3a,b depict the SEM micrographs of the PCL-coated 13-93B3 bioactive glass scaffolds in both the absence and presence of graphene nanopowders, respectively. The thin PCL layer covering the scaffolds can be clearly observed from the images, and the inclusion of graphene nanopowders created additional sites on the surface of the scaffolds. Figure 3 also reveals that, although the applied coating partially occluded pores at the surface, an interconnected open pore network still existed in the scaffolds.

Measurements showed that total porosity of the bare bioactive glass scaffolds in absence of surface coating was ~77%. It declined to 69% with the application of a polymeric coating on the surface of the bioactive glass scaffold. The incorporation of graphene nanopowders at the highest concentration further reduced porosity to 65% (Figure 4).

The results of the XRD examination revealed that the sintered bioactive glass scaffolds under investigation were amorphous, with no evidence of crystallization (Figure 5a). The FTIR-ATR spectra of the bare and PCL-coated bioactive glass scaffolds is given in Figure 5b. As a result of the B–O stretching vibrations of BO_4_ and triangular BO_3_ units, the glasses’ FTIR spectra exhibit two broad and notable bands with maxima at about 940 and 1370 cm^−1^, respectively. The B–O–B bending vibrations of the BO_3_ and BO_4_ groups are responsible for the medium band at 717 cm^−1^ [22]. The absorbance intensity of these groups was weaker in the spectrum of the PCL-coated bioactive sample compared to the uncoated sample. On the other hand, the hydroxyl and ester groups are represented as a peak around 3500 cm^−1^ and a very strong signal at 1750 cm^−1^, respectively [23], in the spectrum of PCL (Figure 5c). Functional groups of PCL were not seen in the IR spectrum of the PCL-coated bioactive glass scaffolds due to the application of a thin polymer layer on the surface of the glass. In the case of graphene nanopowders, because of a mismatch in charge states between carbon atoms, they have few absorption signals [24]. This little difference induces a very small electric dipole, resulting in a highly clear IR spectrum (Figure 5d).

### 3.2. Histopathological Evaluation

Before necropsy, the body weight of all rats was measured at week 0, week 4, and week 8. Week 0’s mean body weight was 305 g, week 4’s was 262 g, and week 8’s was 280 g. Weights did not differ significantly between groups or weeks (*p* > 0.05).

Based on the histopathological examination results of the 4-week groups, the graft material was observed in all animal groups at different ratios. The most significant amount of graft material was detected in group 4. Results also revealed that the absorption rate of the graft material was related to the concentration of graphene nanopowders. As the graphene concentration of the scaffolds decreased, higher absorption behavior was observed. Fibrous tissue proliferation occurred in all groups (Figure 6). Similarly, 8 weeks after implantation, both cartilage and bone formation were observed in all animal groups. Bone formation was significant starting at group 2, with the greatest amount obtained in group 4 (Table 1). Accordingly, new bone area was 41.26 ± 0.71 mm^2^ in group 4, whereas it was 5.28 ± 0.61 mm^2^ for the control group after 8 weeks. It was also observed that the amount of residual graft material increased for samples containing higher ratio of graphene, which may correlate with a change in the degradation rate of bioactive glass-based scaffolds (Figure 7). The residual material area was calculated to be 12.80 ± 1.92 mm^2^ for the 10 wt% graphene-containing scaffold implanted group; however, it was 7.84 ± 0.92 mm^2^ for the control group under the same conditions. In the study, osteoblasts and osteoclasts were counted in a 1.23 mm^2^ area at 400× magnification. Results showed that after 4 and 8 weeks of implantation, both the osteoblast and osteoclast numbers were significantly higher in group 4 compared to the control group.

A Picro-Sirius red staining analysis performed to examine fibrous tissue revealed that while marked fibrous tissue was observed in the control group, it decreased in the graphene-coated scaffold implanted groups. After 4 weeks, a decrease in the amount of connective tissue and an increase in bone formation was observed in group 4. After 8 weeks, a decrease in fibrous tissue was recorded in all groups. While cartilage tissue was observed in the control group, bone formation percentages were significant in graphene-coated scaffold implanted groups. The highest amount of bone formation occurred in group 4 (Figure 8).

### 3.3. Immunohistochemical Findings

#### 3.3.1. BMP-2 Immunohistochemistry Results

It has been reported that bone morphogenetic protein-2 (BMP-2) is a potent osteoinductive cytokine, which is crucial during bone repair and regeneration. In the current study, a BMP-2 immuno-histochemistry examination revealed the existence of a marked expression in 8-week groups in comparison with 4-week groups. Graphene-coated bioactive glass-implanted groups exhibited more significant expression compared to the bare PCL-coated scaffolds implanted in control group rats (Figure 9).

#### 3.3.2. Smooth Muscle Actin Immunohistochemistry Results

For evaluation of neo vascularization, sections were immuno-stained with smooth muscle actin. Newly formed vessels expressed the marker and an increased vascularization was observed in the 8-week group compared to the 4-week group. In addition, an increase in new vessel formation was observed in graphene-coated scaffold-implanted groups compared to the control group (Figure 10).

#### 3.3.3. ALP Immunohistochemistry Results

ALP immunochemistry results revealed an increased expression in the 8-week groups compared to the 4-week groups. Expressions increased depending on the ratio of graphene (Figure 11).

The overall results of the study showed the ameliorative effect of graphene-coated bioactive glass scaffolds on the healing of bone fracture areas. An increase in new bone formation occurred in the graphene-coated graft implanted groups compared to the control group. The healing rate increased proportionally to the increase in graphene concentration. 

## 4. Discussion

In bone tissue engineering and dentistry, bioactive glasses have a wide range of uses [21,25]. Gu et al. examined the ability of porous silicate and borate-based bioactive glass scaffolds to regenerate bone in rat calvarial lesions in vivo in a previous work [9]. At 12 weeks, the volume of new bone formed in the defects implanted with the 13-93 scaffolds was 31%, compared to 20% for the scaffolds with 13-93B3 bioactive glass. The amount of new bone formed in the 13-93 scaffolds was significantly less than in the 13-93B3 scaffolds. Boron liberated from 13-93B3 glass in surrounding defects has been shown to induce bone formation in 13-93 scaffold implanted defect sites [9].

On the other hand, graphene is a carbon-based material with a honeycomb structure one atom thick. It exhibits a distinctive optical, thermal, and mechanical performance and can be manufactured in various structures to provide applications in the field of biomedicine [26,27,28,29].

While graphene increases the mechanical strength of bioactive glass, it also makes it electroactive, and so graphene-coated bioactive glass scaffolds maintain excellent electroactivity [6]. In this way, graphene-coated mesoporous scaffolds cause significant increases in the differentiation of MSCs and cell-to-cell communication.

Graphene-containing, PCL-coated borate-based 13-93B3 bioactive glass scaffolds are candidates for effective graft options in filling segmental defects of long bones. In this study, the effects of graphene nanopowders embedded in a PCL matrix coated on bioactive glass scaffolds were investigated for bone healing after 4 and 8 weeks of implantation in rats. Since rigid fixation could not be provided, it was not possible for the bridging callus to form into a hard callus in 8 weeks. The presence of PCL as a coating on the surface of the bioactive glass scaffolds prevented the direct rapid release of graphene nanopowders to the tissue site after implantation. Without using a polymer layer, an increase in local graphene concentration could be detrimental for the bone cells, since graphene has a two-dimensional structure and its toxicity is concentration-dependent. Furthermore, the presence of a PCL coating may also enhance the mechanical strength of scaffolds, which is crucial for the repair of the bone tissue. The bone remodeling phase begins after formation of a sufficient amount of bone callus (in the first 28 days), and includes the enchondral ossification process (lasting months to years). In other words, in the first 28 days, the bone defect must be knitted with new blood vessels and the bridge callus must be formed [2,30,31,32]. It is known that an adequate bridge callus cannot be formed in segmental bone defects. For this reason, a bioactive scaffold with a porous structure that imitates cancellous bone is needed [16,33]. The mechanical properties and architectural structure of this scaffold must allow for neo-angiogenesis, ionic interaction and mineralization. Because the vascularization process stagnates after the critical first month, enchondral ossification and mineralization cascades end. At this stage, potent growth factors such as BMP, VEGF, and TGF provide differentiation from MSCs to osteocytes and chondrocytes. Their maintenance and regulation are as important as the initiation of osteoinduction [2,31,34,35].

One of the most important parts of bone defect repair is vascularization, as the amount of new bone required to fill the defect is directly related to the vascularization rate. It has been shown in vitro that borate-based bioactive glass PCL composites can increase VEGF and ANG-1 expression of rBMSCs [7,14]. Moreover, it is known that the incorporation of boron into BGs could enhance osteogenesis in vitro [7]. In the current study, borate-based PCL-coated bioactive glasses constituted the control group. In this way, the stimulating effect of different amounts of graphene nanopowders were clearly observed. According to data obtained with the rat femur defect model, borate-based bare BG scaffolds cannot sufficiently support vascularization and osteogenesis in large bone defects. New vessel formation was examined immunohistochemically (smooth muscle actin) and a significant increase in SMA for the graphene-containing groups observed (Table 2, see also Figure 10).

The replacement of the scaffold with new bone tissue is directly related to the degradation time of the scaffold. With the onset of the bone remodeling phase, the scaffold should begin to degrade. Early or late biodegradation negatively affects the healing process, especially in long bones.

Bone ALP, OPN, and OCN levels are the biomarkers of the osteogenesis process, and their levels increase during the mineralization process [36,37]. The hormonal response and mechanical factors limit excessive bone formation. Potent growth factors such as BMP, VEGF, and TGF cannot be synthesized sufficiently in segmental bone defects. For this reason, in the absence of potent growth factors such as BMP, VEGF, and TGF, the remodeling phase is interrupted [2,38,39].

Studies in the literature have reported the effectiveness of bioactive glass scaffolds on bone union with the calvarial bone defect model [40]. However, there are clear differences between long bone healing and flat bone healing [31,40]. Therefore, we think that the defect model preferred in our study will shed more light on orthopedic surgery practices. The major limitation of this study is that only recovery times of 4 and 8 weeks were examined. Further studies on the in vivo response of the bioactive glass-based composite system under investigation can be performed using graphene oxide, using functional groups instead of pristine graphene, and a more hydrophilic polymer coating layer to enhance the cellular interaction.

## 5. Conclusions

Graphene containing (0, 1, 3, 5 and 10 wt%) PCL-coated porous borate-based 13-93B3 bioactive glass scaffolds were fabricated by polymer foam replication and dip coating methods. The prepared scaffolds were evaluated for their capacity to enhance bone formation in rat segmental defects in vivo. Scaffolds containing graphene showed a better capacity to support new bone formation and alkaline phosphatase activity. The number of osteoclasts and osteoblasts were significantly higher in the 10 wt% graphene-coated bioactive glass scaffold-implanted group compared to the control group. Fibrous tissue formation was observed in the control group, whereas it decreased in the graphene-coated scaffold-implanted groups. Furthermore, the presence of graphene-enhanced BMP-2 and SMA levels was increased compared to bare bioactive glass scaffolds. Pristine graphene-coated bioactive glass scaffolds have a high potential to be used in the repair and regeneration of large segmental bone defects.

## Figures and Tables

**Figure 1 polymers-14-03898-f001:**
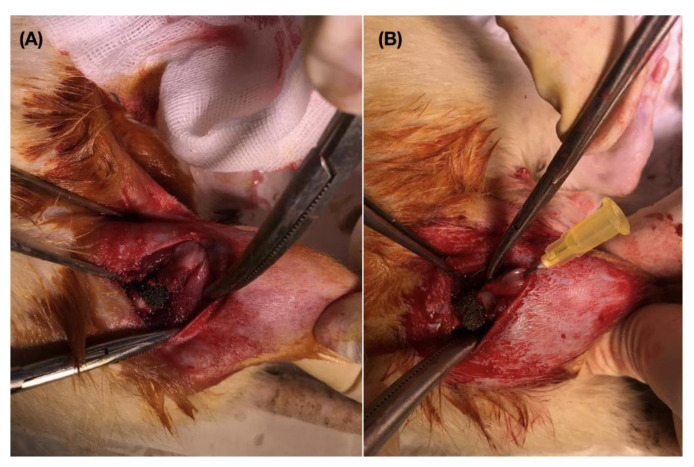
(**A**) 5 mm large femoral defect filled with a porous scaffold. (**B**) Retrograde intramedullary pin fixation.

**Figure 2 polymers-14-03898-f002:**
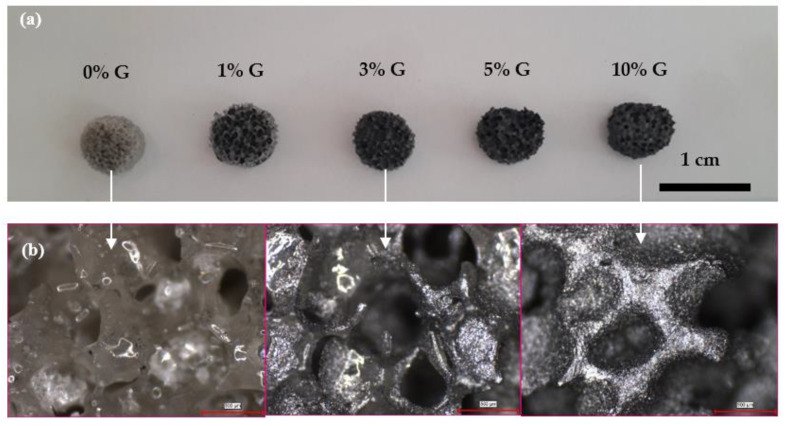
(**a**) Digital and (**b**) optical microscope images of the bioactive glass-based composite scaffolds (PCL-coated, containing 0, 1, 3, 5 and 10 wt% graphene) fabricated in the study, scale bar: 500 µm for optical microscope images.

**Figure 3 polymers-14-03898-f003:**
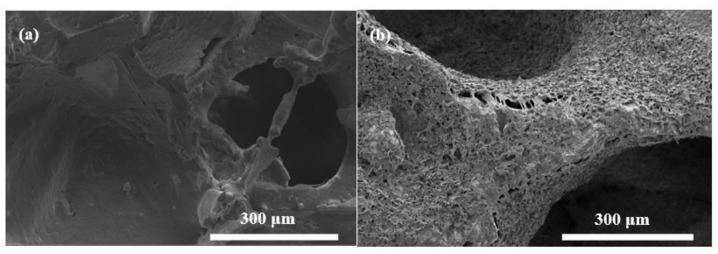
SEM micrographs of the (**a**) bare PCL-coated, (**b**) 10 wt% graphene-containing PCL-coated bioactive glass scaffolds.

**Figure 4 polymers-14-03898-f004:**
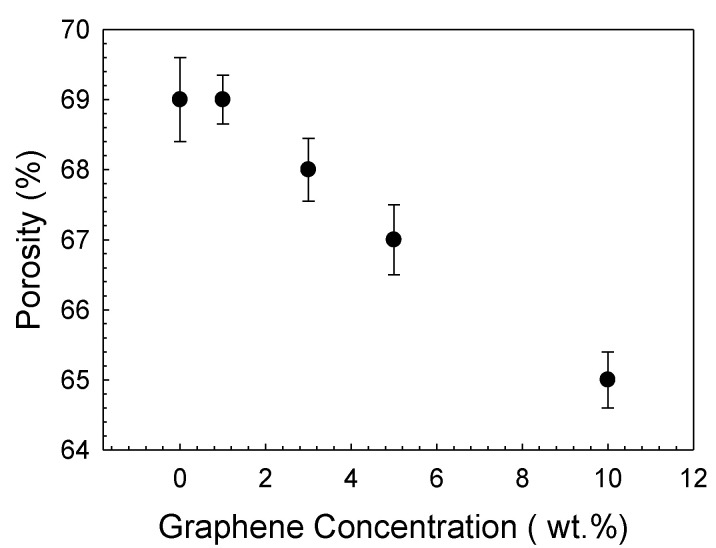
Total porosity of the graphene-containing PCL-coated bioactive glass scaffolds.

**Figure 5 polymers-14-03898-f005:**
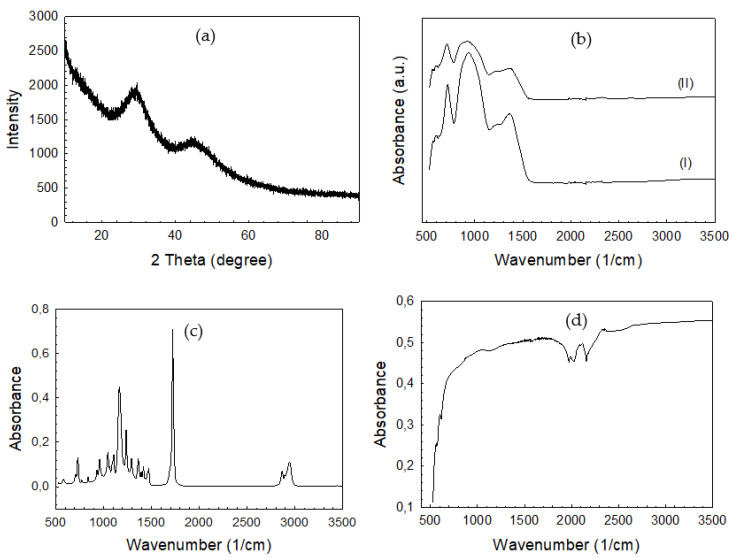
(**a**) XRD pattern of the sintered bare bioactive glass scaffold, (**b**) FTIR-ATR spectra of the bare borate glass (I) and PCL-coated borate glass scaffolds (II), (**c**) FTIR-ATR spectrum of the PCL, (**d**) graphene nanopowders utilized in the study.

**Figure 6 polymers-14-03898-f006:**
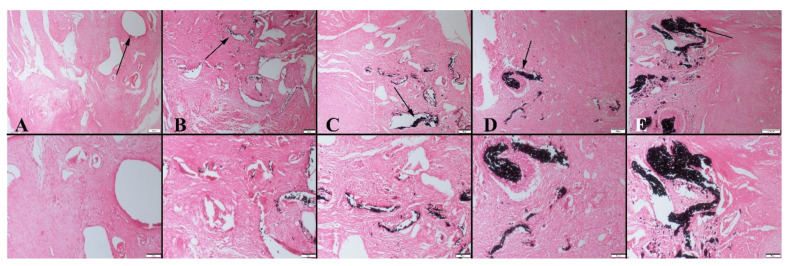
Representative histopathological micrographs of bone fracture areas of(upper row) at 4-week groups, (**A**) Control group, (**B**) Group 1, (**C**) Group 2, (**D**) Group 3 and (**E**) Group 4 lower row higher magnification, graft materials (indicated by arrows), HE, Bars = 200 µm (for upper row) and 100 µm (for lower row).

**Figure 7 polymers-14-03898-f007:**
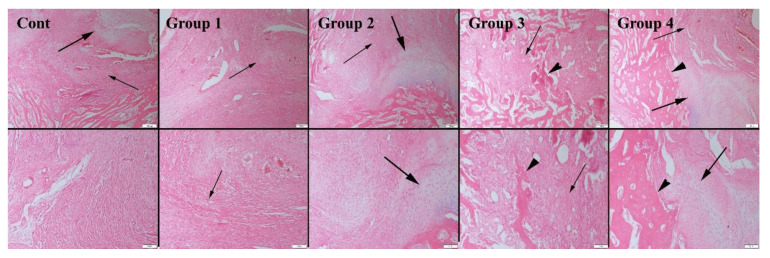
Representative histopathological micrographs of bone fracture areas. (**A**) Control group, (**B**) Group 1, (**C**) Group 2, (**D**) Group 3 and (**E**) Group 4, lower row higher magnification, Fibrous tissue (thin arrows), cartilage (thick arrows) and new bone tissues (arrow heads) (upper row) at 8-week groups, higher magnification (lower row), HE, scale bars = 200 µm (for upper row) and 100 µm (for lower row).

**Figure 8 polymers-14-03898-f008:**
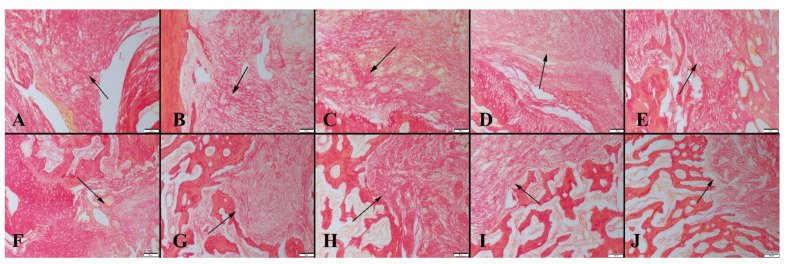
Micrographs showing fibrous tissue formation (arrows) in animal groups obtained via the Picro-Sirius red method. (**A**) Control group, (**B**) Group 1, (**C**) Group 2, (**D**) Group 3 and (**E**) Group 4, for 4-week and (**F**) Control group, (**G**) Group 1, (**H**) Group 2, (**I**) Group 3 and (**J**) Group 4, for 8-week. Decreased fibrous tissue formation is shown in graphene-coated scaffold-implanted groups, scale bar = 100 µm.

**Figure 9 polymers-14-03898-f009:**
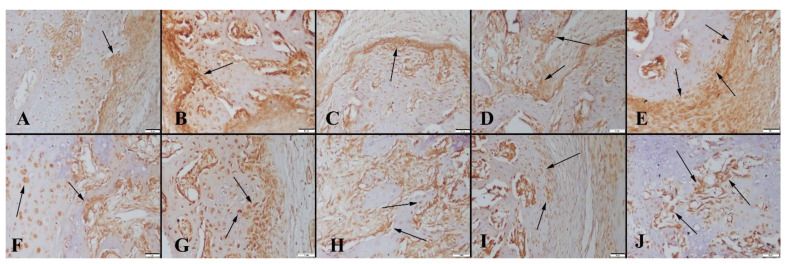
BMP-2 expression between the groups. (**A**) Control group, (**B**) Group 1, (**C**) Group 2, (**D**) Group 3 and (**E**) Group 4, for 4-week and (**F**) Control group, (**G**) Group 1, (**H**) Group 2, (**I**) Group 3 and (**J**) Group 4, for 8-week, arrows indicate immunopositive cells, Streptavidin biotin peroxidase method, scale bars = 50 µm.

**Figure 10 polymers-14-03898-f010:**
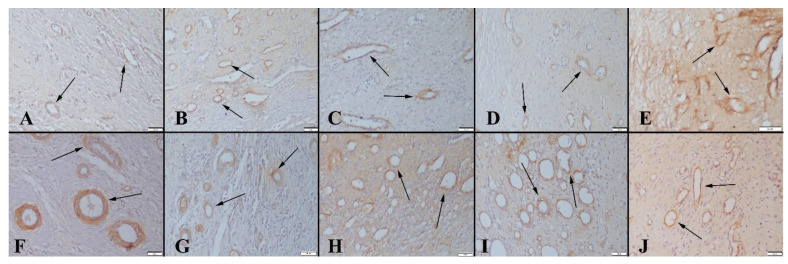
Smooth muscle actin expression among the groups. (**A**) Control group, (**B**) Group 1, (**C**) Group 2, (**D**) Group 3 and (**E**) Group 4, for 4-week and (**F**) Control group, (**G**) Group 1, (**H**) Group 2, (**I**) Group 3 and (**J**) Group 4, for 8-week, arrows indicate immunopositive cells, Streptavidin biotin peroxidase method, scale bars = 50 µm.

**Figure 11 polymers-14-03898-f011:**
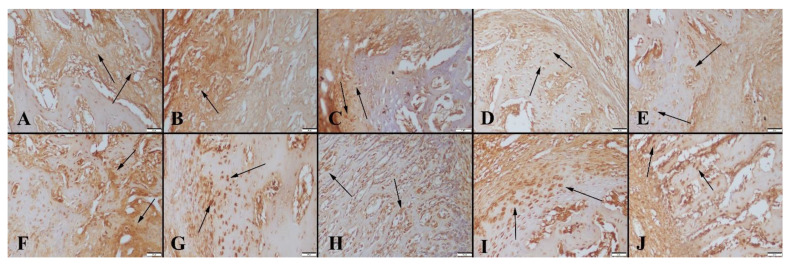
ALP expressions between the groups. (**A**) Control group, (**B**) Group 1, (**C**) Group 2, (**D**) Group 3 and (**E**) Group 4, for 4-week and (**F**) Control group, (**G**) Group 1, (**H**) Group 2, (**I**) Group 3 and (**J**) Group 4, for 8-week, arrows indicate immunopositive cells. Streptavidin biotin peroxidase method, scale bars = 50 µm.

**Table 1 polymers-14-03898-t001:** Statistical analysis of histomorphic data.

		4 Weeks	8 Weeks
New bone area (mm^2^)	Control	2.66 ± 0.38 ^a^	5.28 ± 0.61 ^a^
	Group 1	19.40 ± 2.07 ^b^	23.98 ± 0.91 ^b^
	Group 2	20.24 ± 1.37 ^c^	31.18 ± 1.09 ^c^
	Group 3	25.20 ± 1.48 ^d^	36.00 ± 3.36 ^d^
	Group 4	28.00 ± 1.73 ^e^	41.26 ± 0.71 ^e^
Residual material area (mm^2^)	Control	15.70 ± 1.78 ^a^	7.84 ± 0.92 ^a^
	Group 1	19.40 ± 1.14 ^b^	8.48 ± 0.91 ^a^
	Group 2	18.20 ± 2.28 ^a^	7.48 ± 1.58 ^a^
	Group 3	22.20 ± 1.92 ^c^	10.12 ± 1.19 ^b^
	Group 4	25.40 ± 2.07 ^d^	12.80 ± 1.92 ^c^
Osteoclast number	Control	10.40 ± 0.54 ^a^	14.40 ± 1.14 ^a^
	Group 1	13.80 ± 1.48 ^b^	18.20 ± 1.64 ^b^
	Group 2	19.00 ± 1.87 ^c^	20.00 ± 1.58 ^b^
	Group 3	20.00 ± 1.58 ^c^	24.00 ± 1.58 ^c^
	Group 4	24.00 ± 1.58 ^d^	26.60 ± 2.07 ^d^
Osteoblast number	Control	11.00 ± 1.58 ^a^	20.20 ± 2.28 ^a^
	Group 1	16.20 ± 1.48 ^b^	25.00 ± 1.87 ^b^
	Group 2	17.40 ± 1.14 ^b^	27.60 ± 1.14 ^c^
	Group 3	19.60 ± 0.89 ^c^	28.40 ± 2.96 ^c^
	Group 4	23.60 ± 1.34 ^d^	35.20 ± 0.83 ^d^

Standard deviation of data (SD). A one-way Duncan test was performed for statistical analysis. *p* < 0.001 indicates that differences in the means of groups with different superscript letters in the same column are statistically significant. Differences between groups with the same superscript are statistically insignificant.

**Table 2 polymers-14-03898-t002:** Statistical analysis of immuno-histochemical scores.

	BMP-2	SMA	ALP
**4-week**			
Control	1.20 ± 0.44 ^a^	0.80 ± 0.44 ^a^	1.40 ± 0.54 ^a^
Group 1	2.60 ± 0.54 ^b^	1.40 ± 0.54 ^a^	2.20 ± 0.83 ^b^
Group 2	2.40 ± 0.54 ^b^	1.60 ± 0.89 ^a^	2.00 ± 0.70 ^b^
Group 3	2.40 ± 0.54 ^b^	1.40 ± 0.54 ^a^	1.80 ± 0.83 ^b^
Group 4	2.60 ± 0.54 ^b^	1.80 ± 0.83 ^b^	2.00 ± 1.00 ^b^
**8-week**			
Control	1.00 ± 0.00 ^a^	1.40 ± 0.54 ^a^	1.60 ± 0.54 ^a^
Group 1	1.75 ± 0.50 ^a^	2.20 ± 0.83 ^b^	2.00 ± 0.70 ^b^
Group 2	2.00 ± 0.00 ^b^	2.20 ± 0.44 ^b^	2.20 ± 0.44 ^b^
Group 3	2.00 ± 0.81 ^b^	2.40 ± 0.54 ^b^	2.60 ± 0.54 ^b^
Group 4	2.75 ± 0.50 ^b^	2.40 ± 0.89 ^b^	2.80 ± 0.44 ^c^

Standard deviation (SD). A one-way Duncan test was used in the statistical analysis. *p* < 0.001 denotes a statistical significance of differences in the means of groups with different superscript letters in the same column. Differences between groups with the same superscript are statistically insignificant.

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
