# Peer review of "Bone Healing in Rat Segmental Femur Defects with Graphene-PCL-Coated Borate-Based Bioactive Glass Scaffolds"

_polymers, 2022, doi:10.3390/polym14183898_

Round 1

Reviewer 1 Report

Dear authors, 

The study entitled " Long Bone Healing in Rat Segmental Femur Defects with Gra- 2
phene-PCL-coated Borate-based Bioactive Glass Scaffolds" presents an innovative approach for bone regeneration using a modified biomaterial. The description of the methodology for development and the characterization were well-detailed. The authors showed excellent microscopies and chemical characterization. The in vivo experiment followed all ethical guidelines and it is correctly described. Additionally, the results revealed a superior performance of the new biomaterial against the control. Minor corrections in the text should be addressed to this manuscript has excellent quality for publication. 

1- Abstract: In the last sentence the authors described the word "osteointegration". I do not agree with that sentence. The authors do not investigate osteointegration.

2- The introduction is quite long, the authors can summarize some sentences and paragraphs.

3- In some parts of the manuscripts the authors use the term "long healing". For bone, long healing is considered around 4-6 months. I think the authors should correct this term for " late healing" according to the time applied in this study.

4- IN the discussion section, the authors can provide some sentences with the limitations of this work and future investigations required for this innovative biomaterial.

Author Response

Reviewer 1:

The study entitled " Long Bone Healing in Rat Segmental Femur Defects with Graphene-PCL-coated Borate-based Bioactive Glass Scaffolds" presents an innovative approach for bone regeneration using a modified biomaterial. The description of the methodology for development and the characterization were well-detailed. The authors showed excellent microscopies and chemical characterization. The in vivo experiment followed all ethical guidelines and it is correctly described. Additionally, the results revealed a superior performance of the new biomaterial against the control. Minor corrections in the text should be addressed to this manuscript has excellent quality for publication. 

Comment-1. Abstract: In the last sentence the authors described the word "osteointegration". I do not agree with that sentence. The authors do not investigate osteointegration.

Response-1. We agree with the reviewer. The related word was deleted. 

Comment- 2. The introduction is quite long, the authors can summarize some sentences and paragraphs.

Response-2. Based on the suggestion of the reviewer the introduction section of the manuscript was shorted.

Comment-3. In some parts of the manuscripts the authors use the term "long healing". For bone, long healing is considered around 4-6 months. I think the authors should correct this term for " late healing" according to the time applied in this study.

Response-3. In the manuscript, the term “long” was utilized to describe the bone not for the duration of the healing process. However, to avoid confusion the term length was removed from the title of the manuscript.

Comment-4. IN the discussion section, the authors can provide some sentences with the limitations of this work and future investigations required for this innovative biomaterial.

Response-4. The related discussion was added to the revised version of the manuscript.

Reviewer 2 Report

This manuscript reported " Long Bone Healing in Rat Segmental Femur Defects with Graphene-PCL-coated Borate-based Bioactive Glass Scaffolds ". While the work is of interest, I think a better experimental design could be provided. There are some issues of the manuscript need to be attention. Authors may see the comments as the follow:

  1. Introduction should be focused more on the novelty of this study. Please elaborate it.
  2. Add a schematic figure of your work (methods). This will make it easier for readers to follow contents of manuscript.
  3. Characterization of fabricated scaffolds were missing (Lack of relevant physic/chemical and mechanical analysis).
  4. How to control the porosity of scaffolds? And its mechanical properties?
  5. In-vitro biocompatibility should be performed to demonstrate that the developed scaffolds have no cytotoxicity. The following paper may be useful to choose appropriate biocompatibility test: https://doi.org/10.1002/jbm.b.34636
  6. At Figure 7, as you represent new bone formation, I recommended using Alizarin Red/Alcian Blue staining too.
  7. There is lack of the analysis key gene for bone formation, such as ALP,...
  8. There are many typographical errors that authors need to review. For instance, at line 108: “The glass particles had a median diameter of 10 m”
  9. References are not updated, please remove/replace the outdated references.

Author Response

Reviewer 2

This manuscript reported " Long Bone Healing in Rat Segmental Femur Defects with Graphene-PCL-coated Borate-based Bioactive Glass Scaffolds ". While the work is of interest, I think a better experimental design could be provided. There are some issues of the manuscript need to be attention. Authors may see the comments as the follow:

Comment-1.    Introduction should be focused more on the novelty of this study. Please elaborate it.

Response-1. The contribution of borate-based 13-93B3 scaffolds containing pristine graphene nano-platelets to bone healing in vivo has not been reported yet. Therefore, novelty of the study is to in vivo investigation of the effect graphene (in the bioactive glass matrix) on the new bone formation.

Comment-2.    Add a schematic figure of your work (methods). This will make it easier for readers to follow contents of manuscript.

Response-2. Based on the suggestion of the reviewer a graphical abstract was added to the revised manuscript.

Comment-3.    Characterization of fabricated scaffolds were missing (Lack of relevant physic/chemical and mechanical analysis).

Response-3. Detailed physicochemical and mechanical characterization of the related scaffolds has been reported previously.

Mert Türk, Aylin M. Deliormanlı “Graphene-containing PCL-coated Porous 13-93B3 Bioactive Glass Scaffolds for Bone Regeneration ” Materials Research Express, 5, 4, 045406 (2018). 

Comment-4.    How to control the porosity of scaffolds? And its mechanical properties?

Response-4. In polymer foam replication method there is no need to control the porosity because the porosity the fabricated scaffold will be the same with the template material. A template material (generally PU foam) having a high porosity and pore size will result the fabrication of a glass scaffold with high porosity.

Porosity percentage and the pore size distribution directly influence the mechanical properties of the scaffolds. Further information regarding the related properties of the studied scaffolds can be found in our previously published articles [12-14].

Comment-5.    In-vitro biocompatibility should be performed to demonstrate that the developed scaffolds have no cytotoxicity. The following paper may be useful to choose appropriate biocompatibility test: https://doi.org/10.1002/jbm.b.34636

Response-5.  In vitro biocompatibility (with osteoblastic MC3T3-E1 cells using XTT, live/dead cell assay, ALP activity) of the bioactive glass composite scaffolds under investigation has been reported in our previous article*. 

*Mert Türk, Aylin M. Deliormanlı “Graphene-containing PCL-coated Porous 13-93B3 Bioactive Glass Scaffolds for Bone Regeneration ” Materials Research Express, 5, 4, 045406 (2018). 

Comment-6.    At Figure 7, as you represent new bone formation, I recommended using Alizarin Red/Alcian Blue staining too.

Response-6. Picro Sirius red and HE staining were utilized in the study. Since the study is based on in vivo animal experiments which has been completed months ago currently there is no possibility to conduct further staining procedure due to lack of samples.

For new bone formation and the residual material area, histomor-phometric variables were calculated. At 400x magnification, osteoblasts and osteoclasts were detected in a 1.23 mm2 area . The entire faulty area was in-vestigated in two dimensions and calculations were carried out in five distinct parts of each segment.

Comment-7.    There is lack of the analysis key gene for bone formation, such as ALP,...

Response-7. ALP analysis results can be found in section 3.3.3. ALP Immunohistochemistry results. Figure 11 and Table 2

Comment-8.    There are many typographical errors that authors need to review. For instance, at line 108: “The glass particles had a median diameter of 10 m”.

Response-8. The typographical errors were corrected in the revised manuscript.

Comment-9.   References are not updated, please remove/replace the outdated references.

Response-9. Based on the recommendation of the reviewer some references were replaced with the updated articles published in 2022.

Reviewer 3 Report

  • References [13] and [14] may be the same, it is not clear if they are different or the same.
  • I have some doubts about whether references [15] or [20] are totally relevant, because the text mentions them but the connection with the current work is not so clear to me (some explanation could be added linking those works with the current one).
  • Reference 21 contains errors in the name of one of the authors.

Author Response

Reviewer 3:

Comment-1. References [13] and [14] may be the same, it is not clear if they are different or the same.

Response-1. The related mistake in references were corrected in the revised manuscript.

Comment-2. I have some doubts about whether references [15] or [20] are totally relevant, because the text mentions them but the connection with the current work is not so clear to me (some explanation could be added linking those works with the current one).

Response-2. Reference 15 was corrected. It is based on the preparation of robocast silicate based bioactive glass scaffolds coated with the graphene containing PCL solution. In the current study borate based bioactive glass scaffolds were prepared and coated similarly with the graphene-containing (at the same concentration with the previous work) PCL solution. Because of the use of the graphene at the same concentrations and the PCL for coating in both studies previous one has been cited.   Ref [20] describes the in vivo experiments for graphene-containing PCL scaffolds in the absence of bioactive glass. Therefore, a comparison can be made between two studies.

Comment-3. Reference 21 contains errors in the name of one of the authors.

Response-3. The related mistake in reference 21 was corrected in the revised manuscript.

Round 2

Reviewer 2 Report

Accept in present form